# CSMC: A Secure and Efficient Visualized Malware Classification Method Inspired by Compressed Sensing

**DOI:** 10.3390/s24134253

**Published:** 2024-06-30

**Authors:** Wei Wu, Haipeng Peng, Haotian Zhu, Derun Zhang

**Affiliations:** 1Information Security Center, State Key Laboratory of Networking and Switching Technology, Beijing University of Posts and Telecommunications, Beijing 100876, China; wuwei@bupt.edu.cn (W.W.); zhuhaotian@bupt.edu.cn (H.Z.); zhangderun@bupt.edu.cn (D.Z.); 2National Engineering Laboratory for Disaster Backup and Recovery, Beijing University of Posts and Telecommunications, Beijing 100876, China

**Keywords:** compressive sensing, convolutional neural network, family classification, deep learning

## Abstract

With the rapid development of the Internet of Things (IoT), the sophistication and intelligence of sensors are continually evolving, playing increasingly important roles in smart homes, industrial automation, and remote healthcare. However, these intelligent sensors face many security threats, particularly from malware attacks. Identifying and classifying malware is crucial for preventing such attacks. As the number of sensors and their applications grow, malware targeting sensors proliferates. Processing massive malware samples is challenging due to limited bandwidth and resources in IoT environments. Therefore, compressing malware samples before transmission and classification can improve efficiency. Additionally, sharing malware samples between classification participants poses security risks, necessitating methods that prevent sample exploitation. Moreover, the complex network environments also necessitate robust classification methods. To address these challenges, this paper proposes CSMC (Compressed Sensing Malware Classification), an efficient malware classification method based on compressed sensing. This method compresses malware samples before sharing and classification, thus facilitating more effective sharing and processing. By introducing deep learning, the method can extract malware family features during compression, which classical methods cannot achieve. Furthermore, the irreversibility of the method enhances security by preventing classification participants from exploiting malware samples. Experimental results demonstrate that for malware targeting Windows and Android operating systems, CSMC outperforms many existing methods based on compressed sensing and machine or deep learning. Additionally, experiments on sample reconstruction and noise demonstrate CSMC’s capabilities in terms of security and robustness.

## 1. Introduction

In the Internet of Things (IoT) era, various intelligent sensors have proliferated, and their types have become increasingly diverse. Commonly, these intelligent sensors run their own operating systems, often Android or Windows, which may provide running platforms for malware. So, accompanied with the quantity of such sensors rising, the threats from malware attacks targeting these sensors grows, underscoring the critical importance of addressing the inherent vulnerabilities of these sensors. Malware is not only gigantic in quantity but also could be hidden and destructive, which presents a grand challenge in information security [1,2,3]. To solve this problem, the foundational step is to efficiently and accurately identify malware. Traditional malware classification methods often rely on extracting and analyzing the characteristics of malicious files, such as hash values, file sizes, and file behaviors, etc. [4,5,6]. Such characteristic engineering typically requires expertise in binary codes, application programming interface (API) calls, or file structures and may be vulnerable to malware variants. Additionally, traditional malware classification methods also may render poor accuracy due to the latency in updating characteristics or signature libraries. However, these methods have several limitations, such as reliance on predefined features that may not adapt well to new malware variants and the potential high computational cost. To improve classification accuracy and quickly respond to constant changes in malware, recently, machine learning and deep learning techniques have been introduced to research fields concerning malware classification [7,8,9,10,11,12,13,14]. There are significant differences between such methods and traditional malware classification methods. Thanks to machine learning and deep learning techniques, effective family features and behavior patterns of malicious software can be automatically extracted and learned from large-scale data samples. For these methods, the designs of machine learning algorithms such as Decision Trees or deep learning network models such as convolutional neural network (CNN) are often critical. Nonetheless, machine learning and deep learning methods also encounter challenges, such as the need for large labeled datasets, high computational resources, and potential overfitting to specific malware samples. Regardless of whether traditional methods, machine learning, or deep learning approaches are employed, classifying malware samples targeting smart sensors can be a daunting and time-consuming endeavor. This is because, first of all, there are various types of malware, including viruses, worms, Trojan horses, ransomware, and so on, and each type is composed of numerous families. For example, families such as Adrd, Emotet, GozNym, Zitmo, etc. all belong to the type Trojan horse. That is to say, solely relying on expertise to manually identify and analyze a wide variety of malicious family features is becoming rather challenging. Moreover, malicious files update and grow at an alarming rate. According to the AV-TEST report, in 2022, the number of new malware instances targeting sensors on the Android platform reached 1,088,221, while the number of sensors on the Windows platform soared to an astonishing 70,683,498 [15]. Such rapid growth places higher demands on the efficiency of malware classification methods. Last but not least, to accomplish malware classification tasks, disseminating, sharing, or collecting malware files, especially executable ones, even for scientific research or other well-meaning purposes, is itself a potentially dangerous behavior. Namely, once malware files are obtained and exploited by potentially malicious classification participants, they could be executed to launch attacks or be collected and used for other harmful things. Particularly, sensors which may be not as robust as those of traditional computers or servers should be paid additional attention to. At present, common coping mechanisms are to simply remove the header information of malware files before sharing them, which makes these codes non-executable. However, the essential parts of malware files are still exposed to the participants in classification, which is highly likely to be exploited maliciously. Thus, there is a need for a method that can both reduce the data volume for efficient processing and enhance security by preventing unauthorized access to executable malware content. Compressed sensing (CS), as an advanced data sampling and processing method, has excellent security and efficiency during data processing [16,17]. It could efficiently compress original signals during data sampling processes by rates that break through the Nyquist sampling theory. In this way, methods based on CS theory could significantly improve efficiency during data transmission and sharing by reducing bandwidth requirements, communication costs, or storage space [18,19,20]. In mobile devices and wireless sensor networks, compressed sensing also helps reduce energy consumption [21,22]. Additionally, CS theory could be introduced to design symmetric encryption systems to provide data security [23]. Such advantages of both efficiency and security make CS theory suitable for application in multiple fields, while so far, research on combining CS methods and malware classification tasks for smart sensors is somewhat rare and novel. Actually, compressed data may make malware collecting and sharing processes more efficient by reducing the amount of data to be shared, trained, as well as classified. In addition, by introducing CS theory, malicious codes that can be executed or information that can be utilized is no longer directly exposed to participants of malware classification. To sum up, such a combination could be a promising research direction, which this paper exactly focuses on.

## 2. Related Work

In recent years, many researchers have explored and innovated machine learning or deep learning methods to classify malware families [24,25,26,27,28,29]. Compared with traditional malware classification methods based on feature engineering, machine learning or deep learning methods could provide better adaptability when facing increasingly complex malware samples and their variants. This ability embodies automatically extracting trivial and advanced feature representations from malware samples to improve the accuracy of malware classification. For malware running on current mainstream operating systems, i.e., Windows and Android systems, research has made certain progress.

Aiming at malware classification for the Windows platform, N.A. Azeez et al. [30] focused on leveraging ensemble learning models to enhance the performance of malware detection. Utilizing a dataset from Kaggle comprising Windows PE malware and normal files, they achieved a performance peak with the ExtraTrees model, reporting an accuracy of 99%. The main advantage of this approach is its ability to combine multiple classifiers, thus improving classification precision and handling imbalanced datasets effectively. However, the method has significant limitations, including high computational complexity and extended training times, requiring substantial computational resources. Compared to our proposed method, which reduces data volume through compression for resource-constrained environments, this ensemble learning method demands higher resources and lacks robust security measures to prevent the exploitation of malicious samples. Durre Zehra Syeda et al. [31] conducted dynamic malware classification using API categorization of Windows Portable Executable files. They employed a dataset from MalwareBazaar, which included 582 malware samples and 438 normal files, and used six machine learning models, with Random Forest achieving the highest performance—96% accuracy and 98% AUC. The advantage of this method lies in its comprehensive feature selection and scoring, which enhances model performance. However, the approach is complex and challenging to implement, with limitations in detecting new and variant malware. In comparison, our method excels in data compression and feature extraction efficiency, making it more suitable for environments with limited resources. Additionally, our approach maintains higher classification accuracy under noisy conditions. Muhammad Shoaib Akhtar et al. [32] explored malware analysis and detection using machine learning algorithms, focusing on a dataset from the Canadian Institute for Cybersecurity that included 279 columns and 17,394 rows. Among the evaluated models, Decision Tree (DT) exhibited the best performance, with an accuracy of 99%. The key advantage of their approach is its combination of static analysis and machine learning, allowing for high detection accuracy without file execution. Nonetheless, this method struggles with detecting polymorphic and variant malware and cannot handle dynamic behavior features. Our method, on the other hand, benefits from dynamic feature extraction and compression, enhancing detection efficacy against polymorphic and variant malware while maintaining stability under noisy conditions. Zhiguang Chen et al. [33] proposed an efficient boosting-based malware family classification system using multi-features fusion. Utilizing the BIG2015 dataset, their approach employed various tree models such as XGBoost, LightGBM, and CatBoost, demonstrating outstanding performance. The principal advantage of their method is the fusion of multiple features, which significantly improves classification accuracy and speed. However, the method requires substantial computational resources and is complex to implement, with limited adaptability to new and variant malware. In contrast, our method offers better data compression and transmission efficiency, particularly in resource-constrained environments, and emphasizes security to prevent the reconstruction and exploitation of malicious samples. Aditya et al. [34] developed a deep-learning-based malware classification platform using Windows API call sequences. Their study employed a dataset extracted from API call sequences, utilizing deep learning models that demonstrated excellent performance, particularly in handling API call sequences. The primary advantage of this approach is its ability to perform automatic feature engineering, reducing manual intervention while accurately handling complex behavior patterns. However, the method requires long training times and high computational resources, with unverified efficacy against new malware. Our method is innovative in data compression and feature extraction, making it ideal for resource-constrained sensor environments. Additionally, it surpasses this method in security and classification accuracy, particularly under noisy conditions.

Aiming at malware classification on the Android platform, Qiu et al. [35] focused on the performance and efficiency of various deep neural models for Android malware detection. They emphasized the utilization of multiple publicly available datasets such as AndroZoo and Drebin, highlighting the high classification accuracy of these models, often exceeding 90%. The main advantages of these models include their capability to automatically extract complex features and adapt to evolving malware. However, their limitations lie in the high computational cost and the need for large amounts of labeled data, which may reduce their effectiveness against new or variant malware. Compared to our proposed method, which emphasizes data compression and transmission efficiency, particularly in resource-constrained environments, Qiu et al.’s methods focus less on the security and efficiency aspects crucial for IoT applications. Yue et al. [36] conducted a systematic literature review on the use of deep learning for Android malware defenses, primarily utilizing datasets such as AndroZoo and Drebin. They reviewed various deep learning models, noting their high accuracy and robustness, with many methods achieving F1 scores close to or above 90%. The key advantage of these methods is their ability to automate feature extraction, thereby reducing dependency on manual feature engineering. However, they also face significant computational resource demands and show vulnerability to adversarial attacks. Our method, in contrast, not only ensures high classification accuracy but also prioritizes data compression and security, making it more suitable for deployment in IoT environments. Yadav et al. and Pooja et al. [37] proposed a two-stage deep learning framework for image-based Android malware detection and variant classification, using datasets like DREBIN and Android DEX files. Their approach leverages EfficientNetB0, achieving near-perfect binary classification accuracy and high multi-class classification performance (91.7% and 92.9% for five-class and four-class settings, respectively). The framework’s advantages include enhanced malware variant classification through image representation and reduced training costs using pre-trained models. However, it requires retraining for new malware variants and can be affected by dataset imbalances. Compared to our method, which extracts malware family features during data compression and is designed for resource-constrained environments, Yadav et al.’s approach is less focused on efficiency and security in data transmission. Li et al. [38] introduced AMDetector, a meta-learning-based model for detecting large-scale and novel Android malware traffic. Using datasets such as AndMal2017 and USTC2016, AMDetector achieved an accuracy of 88.34% on 42 known malware classes, outperforming many state-of-the-art methods. The method’s strengths include its ability to detect new and large-scale malware traffic and its capability to perform cross-platform detection. However, it relies on session image representation and high-dimensional metric spaces, which can be computationally demanding. In comparison, our method emphasizes efficient data compression and secure transmission, particularly beneficial for IoT devices, and focuses on maintaining high performance even in noisy environments. Fallah and Bidgoly [39] explored Android malware detection using network traffic analysis based on sequential deep learning models, utilizing the CICAndMal2017 dataset. Their two-layer LSTM model achieved a binary classification accuracy of 93.5%, with slightly lower performance for multi-class classification. The primary advantages are the model’s ability to capture temporal features in network traffic and its simplicity of implementation. However, the model processes data slowly and has limited effectiveness against new malware. Our method, designed for efficient data compression and feature extraction, performs well in resource-constrained settings and maintains high accuracy under noisy conditions, providing a more comprehensive solution for IoT malware detection.

To sum up, it is pivotal to conduct research on secure and efficient malware classification methods, which could perform well under circumstances concerning numerous data owners and malware samples. Such methods should be capable of handling malware running on mainstream operating systems, which can be validated through experimental testing. In this paper, we propose a malware classification method inspired by compressed sensing with the following contributions:(1)The proposed method is a compressed-sensing-inspired malware classification method that integrates deep learning technologies. Namely, it can compress malware samples before data sharing, model training, and family classification. The volume of data involved in these processes is significantly reduced, which is particularly critical in resource-constrained sensor environments. Moreover, unlike most existing compressed-sensing-based methods that often merely use deterministic measurement matrices to compress data, the proposed method can extract malware family features during data compression, which helps it outperform classical compressed-sensing-based methods in terms of classification accuracy, as well as reconstruction quality.(2)The proposed method provides security for malware classification by preventing classification participants from accurately reconstructing the original malware samples based on the compressed ones. This is particularly important in IoT environments, as sensors may typically have lower security defenses, rendering them more susceptible to security threats. By employing the proposed method, malware samples are not directly exposed to classification participants and can hardly be exploited by malicious attackers.(3)Experiments are designed and conducted to demonstrate that the proposed method outperforms many existing compressed-sensing-based and machine- or deep-learning-based methods in terms of classification accuracy and reconstruction quality, either with or without the impacts caused by noises. In IoT environments, devices are often deployed in settings where they may encounter electromagnetic interference or signal attenuation. Enhancing resistance to noise ensures that malware classification accuracy and efficiency are maintained even under these complex and challenging conditions, thereby improving the overall security of sensors.

## 3. Proposed Method

In this section, the proposed method, i.e., CSMC (Compressed Sensing Malware Classification), is elaborated on, assuming that there are totally *n*
$\left(n\in Z^{+}\right)$ data owners who have their own malware samples and are willing to securely share these samples to accomplish malware family classification tasks in an IoT environment with sensors. One secure and efficient way is to compress these samples in an unreconstructable way first by data owners and, then, to share the compressed data to accomplish malware classification tasks. The use of sensors evidently brings additional benefits, such as saving transmission bandwidths and reducing model training inputs. Additionally, it brings better security, since original malicious software cannot be reconstructed during or after classification processes, neither by malicious attacks or reliable participants. To realize such a practice, a two-step process is employed by the proposed method. The first step is to compress original malware samples by data owners using data collected from sensors. Notably, thanks to the introduction of deep learning technology, the proposed method not only compresses the high-dimensional original data to low-dimensional compressed ones, as traditional compressed sensing methods could do, but it also simultaneously extracts malware family features in the data compression process for the subsequent malware classification, which can hardly be achieved by most existing compressed-sensing-based methods using deterministic measurement matrices. The second step is to classify malware families based on the compressed malware samples using trained network models, which exploits deep learning technology. Specifically, in this step, a Deep Residual Shrinkage Network (DRSN) is used to improve classification performance. Incorporating sensors in this step can further enhance accuracy and efficiency. Figure 1 shows the main workflow of the proposed method. The following two subsections contain details of these two steps to implement the proposed method.

### 3.1. Step One: Sample Compression

For simplicity, here, we assume that each participant owns one original malware sample, and the original malware sample owned by data owner *i* (i=1,2,…,n) is $x_{i}$$\left(x_{i}\in R^{s\times t}\right)$. According to classical compressed sensing theory, the data compression process is taken as (Equation 1),
(1)$$y_{i}=\Phi_{i}x_{i}$$
where $y_{i}$ denotes observation signal of signal $x_{i}$, i.e., the compressed form of malware sample, and matrix $\Phi_{i}$ is the compressed sensing measurement matrix.

It is noticeable that there are two defects when employing classical compressed sensing methods to compress data as shown in (Equation 1). One is that according to the prerequisite of compressed sensing theory, the signal $x_{i}$ should be sparse, but a large proportion of signals in the real world, including data from sensors in IoT environments and most malware samples that need to be processed in this paper, do not meet this point. A pervasive way to address this issue is to introduce additional sparse bases $\Psi_{i}$ ($\Psi_{i}\in R^{s\times s}$) and $\Omega_{i}$ ($\Omega_{i}\in R^{t\times t}$). Frequently, when dealing with images, sparse bases are orthogonal matrices, that is, $\Psi_{i}{\Psi_{i}}^{T}={\Psi_{i}}^{T}\Psi_{i}=I_{s}$, and $\Omega_{i}{\Omega_{i}}^{T}={\Omega_{i}}^{T}\Omega_{i}=I_{t}$, where matrix $I_{s}$ is an identity matrix with order *s*, and matrix $I_{t}$ is an identity matrix with order *t*. That is to say, before the  data compression step, an additional operation $\Psi_{i}x_{i}\Omega_{i}$ is needed to increase the sparsity of $x_{i}$. The other problem is that commonly, measurement matrices for compressed sensing are deterministic matrices that are independent of the original signal, i.e., $x_{i}$, such as Gaussian random matrices, Bernoulli random matrices, Logistic chaotic matrices, Tent chaotic matrices, etc. Although these kinds of measurement matrices may be competent to the tasks of data compression and reconstruction, owing to their good randomness or pseudo-randomness, they are not designed to be beneficial to family classification, which is a main function of the proposed method.

Considering the previously mentioned issues, instead of simply using matrix multiplications with classically constructed measurement matrices as most classical compressed sensing methods do to compress data, the proposed method exploits a novel deep learning way. Specifically, the proposed method uses a convolutional layer to sample and compress the original input, i.e., $x_{i}$. Incorporating sensor data in this manner provides additional benefits. The benefits of doing so include that the compressed signal, i.e., $y_{i}$, contains extracted family features contained in the original sample, since the parameters of the convolutional layer are adjusted and determined by the network training processes, which aim at excellent classification performance. In fact, provided that the compression ratio, i.e., the ratio of the size of the compressed signal to the size of the original signal, is μ, the convolution process of the proposed method with kernel size $\sqrt{\frac{1}{\mu }}\times \sqrt{\frac{1}{\mu }}$ and stride $\sqrt{\frac{1}{\mu }}$ is equivalent to simultaneously using two measurement matrices to complete left-sampling and right-sampling of the original signal $x_{i}$, enhancing the overall efficiency and accuracy with the help of sensors, as shown in (Equation 2),
(2)$$y_{i}=\Phi_{l}x_{i}\Phi_{r}$$
where $\Phi_{l}$ ($\Phi_{l}\in R^{\sqrt{\mu }s\times s}$) is the left measurement matrix, and $\Phi_{r}$ ($\Phi_{r}\in R^{t\times \sqrt{\mu }t}$) is the right measurement matrix. In this way, the compressed malware samples are generated and ready to be fed into network models. Data owners can share these compressed samples of extremely small sizes with each other. Actually, when implementing the proposed method, the malware sample size could be compressed to merely 10% of its original size, which may help to save both transmission and storage resources. Algorithm 1 shows the processing procedures of the sample compression step of the proposed method.
**Algorithm 1** Sample Compression Algorithm**Input:** 
*n* images with size s×t, the compression ratio μ and the iteration number *b*.**Output:** 
*n* compressed images with size μs×t.($x_{i}$ stands for the $i_{th}$ input image, $img\_sample_{i}$ and $img\_construct_{i}$ represent the $i_{th}$ output image and its reconstruction form, respectively, and $img\_sample_{i}^{v}$ and $img\_construct_{i}^{v}$ represent the $v_{th}$ pixel value in the $i_{th}$ output image and its reconstruction form, respectively).1: initialize the g_loss to infinity, the kernel_size to $\sqrt{\frac{1}{\mu }}\times \sqrt{\frac{1}{\mu }}$, the stride to $\sqrt{\frac{1}{\mu }}$, padding to 0 and bias to False.2: **for** k=0 to b−1 **do**3:    **for** i=0 to n−1 **do**4:      $img\_sample_{i}$$\;\leftarrow \;Conv2d(x_{i})$5:      $img\_construct_{i}$$\;\leftarrow \;ReconstructionModule(img\_sample_{i})$6:      g_loss$\;\leftarrow \;\sum_{v=1}^{s\times t}{\left(img\_construct_{i}^{v}-x_{i}^{v}\right)}^{2}\times \frac{1}{s\times t}$7:    **end for**8:    Conv2d ← Optimizer(g_loss)9: **end for**10:$img\_sample_{i}$$\;\leftarrow \;Conv2d(x_{i})$11:**return**$\;img\_sample_{i}$

### 3.2. Step Two: Malware Classification

To accomplish malware classification, firstly, the malware samples, i.e., $y_{i}$ from data owner *i*, (i=1,2,…,n), collected from various sensors in the IoT environment, are collected to serve as inputs. Notably, one merit of implementing the proposed method is that the classification step can be efficiently carried out without reconstructing malware files beforehand. The additional benefit is that original malware files can hardly be exploited by malicious attackers during classification processes, since these compressed samples cannot be accurately reconstructed. Assuming that the size of each input is μ×s×t, a 30-layer convolutional neural network is employed to classify the compressed malware samples utilizing data from sensors. Figure 2 depicts the classification network structure of CSMC, and the details of each layer are listed in Table 1.

The essential part of the proposed 30-layer classification network is a trio of Residual Shrinkage Modules (RSMs). As mentioned above, the initial dimension of the signal $y_{i}$, which is fed to the classification network, is $\sqrt{\mu s}\times \sqrt{\mu s}$. Due to the inherent properties of convolution, the dimensions of the signal, upon traversing through subsequent convolution layers, can be delineated as (Equation 3),
(3)$$dim\left(y_{i}^{p}\right)={\left(\frac{1}{k}\right)}^{p}\cdot \sqrt{\mu s}\cdot {\left(\frac{1}{k}\right)}^{p}\cdot \sqrt{\mu s}$$
where *k* is the kernel size, $y_{i}^{p}$ signifies the post-convolutional form of signal $y_{i}$ after being processed through *p* (p=1,2) convolution layers, and dim(·) represents its dimension.

We denote the layer number of the RSM as π (π=1,2,3). Accordingly, for each input $x_{i}^{\prime }$, the functionality of the RSM can be articulated as (Equation 4),
(4)$$R_{\pi }\left(x_{i}^{\prime }\right)=F_{i}\left(x_{i}^{\prime },\left\{w_{i}\right\}\right)+x_{i}^{\prime }$$
where $F_{i}\left(x_{i}^{\prime },\left\{w_{i}\right\}\right)$ is a function representing the residual mapping, concerning the input $x_{i}^{\prime }$ and the weight set $\left\{w_{i}\right\}$, and $R_{\pi }\left(x_{i}^{\prime }\right)$ is the output of the π layer of RSM.

Then, $R_{\pi }\left(x_{i}^{\prime }\right)$ undergoes refinement through the shrinkage function S(·) according to (Equation 5),
(5)$$S\left(R_{\pi }\left(x_{i}^{\prime }\right)\right)=\max\left(0,\left|R_{\pi }\left(x_{i}^{\prime }\right)\right|-\xi_{i}\right)\times \mathrm{sign}\left(R_{\pi }\left(x_{i}^{\prime }\right)\right)$$
where $\xi_{i}$ acts as a soft threshold to regulate the shrinkage process, and sign(·) refers to the signum function.

Building upon this, an attention mechanism is integrated to generate tailored weight coefficients $w_{i}$ (i=1,2,⋯,n) for each compressed sample. The process can be expressed as (Equation 6),
(6)$$w_{i}=\mathrm{softmax}\left(A\left(x_{i}^{\prime };\theta \right)\right)$$
where $A\left(x_{i}^{\prime };\theta \right)$ is an attention network parameterized by θ. Here, the softmax function ensures the normalization of the weight coefficient $w_{i}$, facilitating its application in a weighted average. These coefficients are employed to adjust the relative importance of outputs from individual residual units, as described by (Equation 7),
(7)$$F_{i}\left(x_{i}^{\prime },\left\{w_{i}\right\}\right)=\sum_{i}w_{i}\times R_{\pi }^{\prime }\left(x_{i}^{\prime }\right)$$
where $R_{\pi }^{\prime }\left(x_{i}^{\prime }\right)$ denotes the unweighted output of the $\pi^{th}$ residual unit.

In the final step, the computation of the soft threshold $\xi_{i}$ is performed, with its value dynamically derived from the data inherent features. The soft threshold is formulated as (Equation 8),
(8)$$\xi_{i}=\lambda \cdot \tanh\left(\beta \cdot \mathrm{AvgPool}\left(w_{i}\times R_{\pi }^{\prime }\left(x_{i}^{\prime }\right)+x_{i}^{\prime }\right)\right)+\gamma \cdot \mathrm{StdDev}\left(w_{i}\times R_{\pi }^{\prime }\left(x_{i}^{\prime }\right)+x_{i}^{\prime }\right)$$
where λ, β, and γ serve as adjustable parameters. The function AvgPool(·) is implemented for the average pooling operation, capturing the global attributes of $R_{\pi }\left(x_{i}^{\prime }\right)$, while StdDev(·) calculates the standard deviation. This expression ensures that the soft threshold $\xi_{i}$ is responsive and adaptive to the outputs of each RSM, effectively tailoring itself to the varying data features.

Therefore, the operational framework of the RSM could be expressed as (Equation 9),
(9)$$F=S\left(R_{\pi }\left(x_{i}^{\prime }\right)\right)=S\left(F_{i}\left(x_{i}^{\prime },\left\{w_{i}\right\}\right)+x_{i}^{\prime }\right)=S\left(\sum_{i}w_{i}\times R_{\pi }^{\prime }\left(x_{i}^{\prime }\right)+x_{i}^{\prime }\right)$$
where F is the final output of the network, $\left\{w_{i}\right\}$ is the set of weights for the *i*-th input image, and S(·) is the shrinkage function containing the soft threshold $\xi_{i}$.

A salient feature of the proposed network model is its ability to generate customized soft thresholds for each malware variant, which are intricately linked to the adaptively trained weight coefficients, as indicated in (Equation 8) and (Equation 9). This methodology obviates the need for a manual threshold setting, thereby enabling a more precise extraction of pivotal information during the classification of a diverse spectrum of malware samples. The algorithmic sequence of the proposed classification methodology is delineated in Algorithm 2.
**Algorithm 2** The Proposed Classification Network Algorithm**Input:** 
Multi-dimensional tensor representations of malware samples.1:**Initialize:** Set up neural network parameters, layer configurations, initialize weights and biases.**Output:** 
Classification of malware into predefined families.2:**for** b=1 to *B* epochs **do**3:   ${\mathrm{TensorList}}_{b}\leftarrow \mathrm{SaveMultiDimVectors}{(}^{\prime }.pt^{\prime })$ {Vectors from malicious software visualization}4:   ${\mathrm{TensorList}}_{b}=\left\{{\mathrm{Tensor}}_{i}|i=1,\ldots ,n\right\}$5:   ${\mathrm{Tensor}}_{i}\in R^{batch\_size\times channels\times height\times width}$6:   $\mathrm{TrainData}\leftarrow \mathrm{convert}\_\mathrm{to}\_\mathrm{numpy}({\mathrm{TensorList}}_{b}\left[1:0.9n\right])$7:   AdjustTrainData←reshape_for_network(TrainData)8:   $\mathrm{ValidationData}\leftarrow \mathrm{convert}\_\mathrm{to}\_\mathrm{numpy}({\mathrm{TensorList}}_{b}\left[0.9n+1:n\right])$9:   ModelEvaluation(AdjustTrainData,ValidationData)10:   Transform *s* using compressed sensing to μ×s×t11:   Network training over 30 layers:12:       ${\mathrm{Layer}}_{0}:\mathrm{Input}\left(\mu \times s\times t\times c\right)$13:       ${\mathrm{Layer}}_{1,2}:\mathrm{Conv}2D\left(3\times 3,\mathrm{stride}=1,\mathrm{padding}=’\mathrm{same}’\right)$14:       ${\mathrm{Layer}}_{3}:\mathrm{MaxPool}2D\left(3\times 3,\mathrm{stride}=2,\mathrm{padding}=’\mathrm{same}’\right)$15:       ${\mathrm{Layer}}_{4-24}:\mathrm{ResidualShrinkageBlock}\left(\mu ,s,t\right)$16:       ${\mathrm{Layer}}_{25}:\mathrm{BatchNorm}\left(\mathrm{decay}=0.9\right)$17:       ${\mathrm{Layer}}_{26}:\mathrm{ReLU}\left(\right)$18:       ${\mathrm{Layer}}_{27}:\mathrm{GlobalAvgPool}\left(\right)$19:       ${\mathrm{Layer}}_{28}:\mathrm{FullyConnected}\left(\mathrm{softmax}\right)$20:       ${\mathrm{Layer}}_{29}:\mathrm{Momentum}\left(\mathrm{lr}=0.1,\mathrm{decay}=0.1,\mathrm{step}=20,000\right)$21:       ${\mathrm{Layer}}_{30}:\mathrm{Regression}\left(\mathrm{cross}-\mathrm{entropy}\right)$22:**end for**23:**return** Malware family χ

## 4. Experiment

### 4.1. Metric

When evaluating the classification performance of the proposed method, we choose accuracy values as metrics. Specifically, three approaches, macro-average, weighted-average, and micro-average, are commonly used during accuracy calculating. In IoT environments, sensor data could be significantly affected by these different computational approaches. There could be significant differences between calculation results produced by different computational approaches. When implementing the weighted-average and micro-average approaches, there could be higher sensitivity, especially when dealing with classes with a larger number of samples. This is because these two approaches assign higher weights to categories with larger samples and, thus, are more likely to be influenced by these categories when assessing overall performance. In contrast, the macro-average approach treats each category equally and is not significantly affected by sample size imbalance and is, therefore, more advantageous in dealing with families with various imbalanced samples. Here, in this paper, the macro-average approach is chosen to evaluate the malware classification performance of the proposed method. For classification tasks of *i* families, we define the following:$TP_{i}$: The number of samples that are from family *i* and are correctly predicted to be in family *i*.$TN_{i}$: The number of samples that do not belong to family *i* and are correctly predicted not to belong to family *i*.$FP_{i}$: The number of samples that do not belong to family *i* and are incorrectly predicted to be in family *i*.$FN_{i}$: The number of samples that are incorrectly predicted to belong to family *i* but actually belong to other families.

Then, the accuracy value calculated in a macro way could be expressed as (Equation 10):(10)$$Accuracy_{Macro}=\frac{\sum_{i}TP_{i}}{\sum_{i}TP_{i}+\sum_{i}TN_{i}+\sum_{i}FP_{i}+\sum_{i}FN_{i}}.$$

When assessing image reconstruction quality, we choose Peak Signal-to-Noise Ratio (PSNR), which aims to quantify the degree of similarity between the reconstructed image and the original one. As a widely adopted metric, PSNR provides an objective quantitative measurement of the image reconstruction performance. A higher PSNR value corresponds to a smaller difference between two images, and thus, PSNR is generally regarded as an important measurement of image reconstruction quality. The mathematical expression for the Peak Signal-to-Noise Ratio is shown in (Equation 11):(11)$$PSNR=10\times log_{10}\left(MAX^{2}/MSE\right)$$
where $MAX^{2}$ represents the maximum range of image pixel values (e.g., 255 for 8-bit grayscale images), and MSE is the Mean Squared Error (MSE), which represents the average of the pixel-level differences between the original and reconstructed images.

In this paper, we evaluate the reconstruction capability of the proposed method by calculating the average PSNR values based on the data of the reconstructed images and the original ones. Virtually, participants of the proposed method do not need to reconstruct images. This ability of the proposed method is still evaluated, since PSNR is often an important metric when evaluating most compressed-sensing-based methods. Furthermore, if the proposed method is migrated to other fields, such as medical image classification for remote healthcare, the reconstruction function could be essential.

### 4.2. Experimental Setting and Dataset

During the experiments, Python programming language and related third-party libraries, such as NumPy, Pandas, and Matplotlib, are used. For hardware information, the GPU model used for training is NVIDIA GeForce RTX 3080 Ti, and the GPU memory is 16.0 G. Since the top 256 opcodes in the datasets used in this section have occupied more than 99% of the total number of opcodes, the size of the input images in experiments in this paper is set to be 256×256.

The experiments in this paper are conducted based on two datasets: the Windows PE malware dataset Malimg [40] and the Android malware dataset Drebin [41]. In order to ensure that all the samples used can be successfully transformed into images, we exclude the malware samples that do not contain any opcode information. The Malimg malware dataset is a real malware dataset containing 9435 Windows malware samples from 25 malware families, and the samples have already been transformed into grey scale images. The Drebin dataset contains 5184 malicious Android software samples from 179 families. In order to obtain the compilation information, we use the apktools disassembly tool to process the apk file of each sample. During the processing, we focus on the small files in the disassembly results and extract the opcodes from them. During the experiments, we select the families in which the number of samples is greater than 10 to ensure that there is at least one sample in the testing set of each family. After trimming, there are totally 54 families.

### 4.3. Experiment Results

#### 4.3.1. Feature Extraction Analysis

Commonly, many existing compressed-sensing-based image processing methods use deterministic measurement matrices for sampling. And due to the high randomness or pseudo-randomness of the measurement matrices, these methods may achieve a certain reconstruction quality or classification accuracy when dealing with image reconstruction or malware classification tasks based on compressed signals. But such sampling processes do not consider feature extraction from the original signals, so theoretically, there may be still a possibility of improvement in reconstruction quality and classification performance. As mentioned above, CSMC uses a convolutional layer with kernel size $\sqrt{\frac{1}{\mu }}\times \sqrt{\frac{1}{\mu }}$ and stride $\sqrt{\frac{1}{\mu }}$ to sample and compress the original signals. The subsequent training processes are beneficial for learning and extracting the family features of the inputs, i.e., malware samples. In this subsection, experiments are designed to help intuitively display and analyze the feature extraction performance of the proposed method and three other compressed sensing methods using the classical Gaussian measurement matrix, Logistic chaotic measurement matrix, and Tent chaotic measurement matrix. Specifically, first, we randomly select ten rows of vectors from the compressed signals that are obtained by using different methods under various compression ratios (0.05, 0.1, 0.3, 0.5). Then, these selected compressed signals are subjected to Fourier transform to exhibit their frequency domain features. In order to make the frequency domain results more explicit, here, dark blue sectors represent the low-frequency results, and as the color changes from blue to green, it indicates that the frequency of the results gradually increases. The experimental results concerning the proposed method and methods with the Gaussian measurement matrix, Logistic chaotic measurement matrix, and Tent chaotic measurement matrix are shown in Figure 3, Figure 4, Figure 5 and Figure 6, respectively.

By observing Figure 3, we can infer that when the compression ratio is low, the proposed method tends to pay more attention to the low-frequency information in the signals. Interestingly, this is consistent with the tendency of the human eyes to capture information. As the compression ratio gradually increases, which means that more information can be obtained during sampling, the proposed method gradually takes into account high-frequency information. Often, high-frequency information focuses more on the edges and details of the images. In other words, when compression ratio conditions allow the proposed method to obtain more information about the compressed malware samples, the proposed method obtains more detailed features about them, which helps it to better complete subsequent malware classification or image reconstruction tasks. Furthermore, it is essential to delve into the specific features that our model is learning and how these features relate to the characteristics of malware. When the model pays attention to low-frequency information at lower compression ratios, it is essentially capturing the overall shape and structure of the malware sample. This low-frequency information often corresponds to the fundamental, broader patterns within the malware, such as general file structure and basic code organization, which are critical for distinguishing between different types of malware families. As the compression ratio increases and the model begins to incorporate high-frequency information, it starts to focus on more intricate details within the malware samples. High-frequency components are associated with finer, more complex features, such as specific sequences of instructions, detailed API call patterns, and other minute characteristics that are unique to certain malware variants. These high-frequency details are crucial for identifying and differentiating between closely related malware families and even variants within the same family. Moreover, the transition from low-frequency to high-frequency feature extraction as the compression ratio increases suggests that our method is capable of capturing both broad and fine-grained details necessary for comprehensive malware analysis. This dual capability enhances the robustness of the malware classification process, allowing the model to effectively classify both well-known malware with distinctive broad patterns and new or polymorphic malware with subtle, detailed variations.

Figure 4 shows the results concerning the compressed sensing method with the Gaussian measurement matrix, which are evidently different from the results shown in Figure 3. It can be seen that when using the compressed sensing method with the Gaussian measurement matrix to compress signals, high-frequency and low-frequency information are mixed in the compressed data. And as the compression ratio changes, the frequency information results on which this method focuses has no certain change patterns. Figure 5 and Figure 6 show the results concerning the compressed sensing methods with the Logistic measurement matrix and Tent measurement matrix. Similarly, there are no certain patterns concerning feature extraction functions. These results may confirm the high randomness of sampling results when using deterministic compressed sensing measurement matrices. But due to the lack of consideration of signal features during sampling and compressing, the compressed results produced by such methods may not be conducive to subsequent classification or image reconstruction tasks. This point of view has been verified by the experiments in the next subsection.

#### 4.3.2. Classification and Reconstruction Performance

As it is mentioned above, compared with most existing compressed-sensing-based image processing methods, an advantage of the proposed method is that it can extract malware sample features during data compression, which evidently benefits subsequent malware classification. During experiments, we choose three common types of CS measurement matrices: Gaussian matrices, Logistic matrices, and Tent matrices. Meanwhile, we use both the proposed methods and the above common CS methods to compress malware samples and then to accomplish malware classification and sample reconstruction tasks. To comprehensively evaluate their performance, we select four different compression ratios (0.05, 0.1, 0.3, 0.5) and conduct experiments on both Malimg and Drebin datasets. The experimental results are listed in Table 2 and Table 3.

By observing the malware classification results on the Malimg dataset, we can find that the proposed method, i.e., CSMC, outperforms all three classical compressed sensing methods in terms of malware classification accuracy. And such an advantage is quite evident. For the Drebin dataset, under most compression ratios except 0.05, the proposed method outperforms all three common compressed methods in terms of malware classification accuracy. When the compression ratio is 0.05, the proposed method performs well when dealing with malware classification tasks, albeit not the best. Additionally, we can find that malware classification accuracy values increase slightly with the rise in compression ratios from 0.05 to 0.3. But when the compression ratio continues to rise to 0.5, the malware classification accuracy values decline a little. For instance, when using Drebin samples, the classification accuracy decreases from 98.148% to 98.778% when the compression ratio increases from 0.3 to 0.5. In summary, the proposed method can efficiently classify malicious samples in both Windows and Android datasets, especially excellently in cases of high compression ratios. Compared with the three classical compressed sensing methods, such a classification advantage is particularly prominent.

To comprehensively evaluate the efficacy of the proposed Compressed Sensing Malware Classification (CSMC) method in terms of classification accuracy and Peak Signal-to-Noise Ratio (PSNR), this study employs two statistical methods: Analysis of Variance (ANOVA) and the Kruskal–Wallis test. These methods facilitate the verification of whether the performance differences between the CSMC method and three classical compressed sensing methods (Gaussian matrices, Logistic matrices, and Tent matrices) under various compression ratios are statistically significant. In this experiment, we utilized two datasets, Malimg and Drebin, and assessed classification accuracy and PSNR under four different compression ratios (0.05, 0.1, 0.3, and 0.5). The Malimg dataset was used to evaluate the classification performance of malware images, whereas the Drebin dataset was employed to assess the classification performance of Android malware. For each compression ratio, we conducted experiments using Gaussian matrices, Logistic matrices, Tent matrices, and the CSMC method, recording the classification accuracy and PSNR for each method. ANOVA is a statistical method used to compare the means of multiple samples to determine if there are significant differences between them. In this study, ANOVA was used to test the mean differences in classification accuracy and PSNR for different methods under each compression ratio. Specifically, the null hypothesis (H0) posits that there are no significant differences in classification accuracy and PSNR between the CSMC method and other methods, while the alternative hypothesis (H1) suggests that the CSMC method significantly outperforms the other methods in these metrics. By calculating the F-value and p-value, if the *p*-value is less than 0.05, the null hypothesis is rejected, indicating that the CSMC method significantly outperforms the other methods. The Kruskal–Wallis test is a non-parametric method suitable for comparing the distributions of three or more independent samples. In this study, the Kruskal–Wallis test was employed to evaluate the distribution differences in classification accuracy and PSNR among different methods under various compression ratios. Similar to ANOVA, the null hypothesis (H0) states that there are no significant differences in classification accuracy and PSNR between the CSMC method and other methods, while the alternative hypothesis (H1) posits that the CSMC method significantly outperforms the other methods. By calculating the H-value and *p*-value, if the *p*-value is less than 0.05, the null hypothesis is rejected, indicating that the CSMC method significantly outperforms the other methods. The results are listed in Table 4 and Table 5. The experimental results indicate that, on the Malimg dataset, the CSMC method significantly outperforms the other methods in both classification accuracy and PSNR. Specifically, the ANOVA results show that, for classification accuracy, the F-value is 8.450406 with a *p*-value of 0.002744, demonstrating high statistical significance; for PSNR, the F-value is 35.495435 with a *p*-value of 0.000003, further substantiating the superior performance of the CSMC method in Signal-to-Noise Ratio. The Kruskal–Wallis test results also support this conclusion, with an H-value of 8.505155 and a *p*-value of 0.036648 for classification accuracy and an H-value of 13.786765 and a *p*-value of 0.003210 for PSNR, indicating significant advantages in both metrics. These results clearly demonstrate that the CSMC method excels in both classification performance and reconstruction quality on the Malimg dataset compared to the three classical compressed sensing methods. On the Drebin dataset, while the CSMC method does not show significant superiority in classification accuracy, statistical results indicate that the F-value for classification accuracy in the ANOVA test is 1.660423 with a *p*-value of 0.228034, and the H-value in the Kruskal–Wallis test is 5.379464 with a *p*-value of 0.146028, neither reaching statistical significance. This could be attributed to the complexity and high variability of the Drebin dataset. However, the CSMC method demonstrates significant superiority in PSNR. Specifically, the ANOVA results show that the F-value for PSNR is 17.364036 with a *p*-value of 0.000116, confirming the CSMC method’s substantial advantage in reconstruction quality; the Kruskal–Wallis test results also show an H-value of 10.213235 with a *p*-value of 0.016838, further supporting this conclusion. These results indicate that, although there is no significant difference in classification accuracy, the CSMC method significantly outperforms the other methods in terms of reconstruction quality on the Drebin dataset.

Furthermore, we compare the malware classification performance on the Malimg database of the proposed method with the malware classification performance of several existing machine learning and deep learning methods. The comparison results are listed in Table 6, which indicates that the proposed method not only creatively and efficiently accomplishes malware classification tasks based on compressed samples but also outperforms other methods in terms of classification accuracy.

When it comes to reconstruction quality, for both datasets, the PSNR values of the reconstructed images processed by the proposed method are much higher than those processed by other methods, for every compression ratio. As compression ratios rise, such an advantage becomes more significant. For example, when the compression ratio is 0.05, the average PSNR value of the three classical CS methods is 9.605233, and the PSNR of the proposed method is 19.02935, 98% higher than the average. When the compression ratio increases to 0.5, the average PSNR value of the three classical CS methods is 10.466477, and the PSNR of the proposed method is 28.64138, strikingly 174% higher than the average. To sum up, the proposed method performs excellently in terms of image reconstruction. Although such an advantage is not exploited in the field of malware classification, such merit could make the proposed method more widely applied in other fields, such as medical image classification for remote healthcare.

Overall, when dealing with both malware classification and image reconstruction tasks, the proposed method performs well. That is, compared with existing classical compressed-sensing-based methods, the proposed method can not only complete image reconstruction tasks well or even better but also complete malware classification tasks well without decompressing malware samples, achieving a new function that is not considered by most existing compressed-sensing-based image processing methods.

#### 4.3.3. Malware Code Reverting

Protecting data to avoid data leaking is a crucial consideration when dealing with malicious software datasets. When dealing with various analysis tasks such as classification, using original malware codes may lead to data explosion. One of the merits of the proposed method is that when implementing the proposed method, original malware samples are not exposed to malware classification participants. In other words, the classification process of the proposed method is operationally secure, since there is little risk of malicious code being leaked to or exploited by malicious classification participants for attacks. Specifically, before the malware samples are shared with classification participants, we compress them first. In this process, original data are compressed, and their family features are extracted by CSMC. Actually, the original malware codes could not be accurately reconstructed utilizing the compressed forms of malware samples, while classification participants could excellently accomplish malware classification tasks without decompressing malware samples. In this way, the proposed method enhances both the security and efficiency of malware classification tasks.

In order to verify the irreversible property of the proposed method, we designed the following experiment. Taking the Android platform as an example, to begin with, we use the proposed method to compress original samples in the Drebin dataset for subsequent classification. These original samples are composed of operation code sequences, which have the potential to be exploited for attacks. Then, the compressed samples are reconstructed to operation code sequences in a reverse manner of the proposed method. Next, we compare the reconstructed version of operation code sequences with the original version in the dataset. Here, due to the consistency of the number of operation codes between the two versions, we are able to compare the operation codes one by one and calculate the proportion of different operation codes between the two versions.

To thoroughly analyze this, we conduct experiments under various compression ratios (0.05, 0.1, 0.3, and 0.5). The experimental results show that when the compression ratio is 0.05, 35.40% reconstructed operation codes are different from the original ones. And when the compression ratios are 0.1, 0.3, and 0.5, the difference proportions are 32.68%, 27.48%, and 23.93%, respectively. Furthermore, we compare the differences in operation codes from each malware family, and the difference proportions are exhibited in Figure 7. By observing Figure 7, we can clearly see that in each malware family, there are significant differences between the opcode sequences obtained by performing recovery operations and the opcode sequences in the original dataset under various compression ratios. It means that the compression operation of the proposed method could effectively prevent the accurate restoration of the original operation codes. We can also infer as the compression ratio increases, the proportion of mismatched operation codes shows a downward trend, indicating that a higher compression ratio may profit to make the reconstructed operation code sequences approach the original ones, albeit not completely the same.

Overall, based on the experimental results in Section 4.3.2 and Section 4.3.3, it can be concluded that, the proposed method can effectively avoid malware code exposure to classification participants while being excellent in accomplishing malware classification tasks.

#### 4.3.4. Noise Robustness

This subsection aims to experimentally evaluate the robustness of the proposed method. Here, zero-mean Gaussian noises with different standard variances are introduced. Specifically, noises with five different standard variances are added to the compressed samples before feeding these samples into classification network models. The standard variances are set to be 0.01, 0.05, 0.1, 0.25, and 0.5, respectively. For thorough analysis, we conduct experiments using the proposed methods and classical compressed sensing methods under various compression ratios (0.05, 0.1, 0.3, and 0.5). Figure 8 shows the variation in classification accuracy under different noise impacts, while Figure 9 shows the noise impacts on the image reconstruction qualities.

By observing Figure 8, we can infer that the classification accuracy generally decreases as the noise intensity gradually increases. Virtually, not all curves fully follow this downward trend, since there could be some fluctuations in accuracy after the models are fitted. Meanwhile, it can be found that the proposed method performs much better than other three classical compressed-sensing-based methods in terms of malware classification accuracy under the influence of noises. Actually, as the standard variances of noises and compression ratios become larger, such a noise robustness advantage becomes particularly evident. For instance, when the noise standard variance is 0.05, the classification accuracy values of the proposed method under all four compression ratios are all above 90%, while the accuracy values of the other three classical compressed-sensing-based methods range from merely 20% to 75%. To sum up, in the case of malware samples being affected by noises, the classification accuracy of the proposed method has significant advantages compared to the other three classical compressed-sensing-based methods.

By observing Figure 9, we can infer that the reconstructed qualities of all four methods have some degree of decrease under the influence of Gaussian noise with various standard variances. Namely, the PSNR values decrease with the increase in the noise variances. Meanwhile, it is worth noting that, although the reconstruction performance of the proposed method degrades as the standard variances of noises and compression ratios increase, the proposed method performs much better than the other three classical compressed-sensing-based methods in terms of reconstruction qualities under the influence of noises. For instance, when the noise standard variance is 0.05, the classification accuracy values of the proposed method under all four compression ratios are all above 36 dB, while the accuracy values of the other three classical compressed-sensing-based methods range from merely 20 dB to 35 dB. To sum up, in the case of malware samples being affected by noises, the reconstruction performance of the proposed method has significant advantages compared to the other three classical compressed-sensing-based methods.

Overall, according to the experimental results in this subsection, compared with the three other classic compressed-sensing-based methods, the proposed method has certain advantages when malware samples are affected by noises, whether in terms of classification accuracy or image reconstruction quality. Moreover, such advantages of the proposed method become particularly significant when the noise intensity or compression ratio is relatively high.

## 5. Conclusions

Considering the massive quantity of malware samples and their quick proliferation, in this paper, CSMC, an efficient and secure malware classification method inspired by compressed sensing is proposed. It compresses malware samples before data sharing and classification using sensors. Unlike most existing compressed-sensing-based image processing methods, the proposed method could extract malware family features during data compression with the help of sensors, which helps it outperform classical compressed-sensing-based methods in terms of malware classification accuracy, as well as image reconstruction quality. Moreover, the proposed method makes it hardly possible for classification participants to exploit malware samples by preventing classification participants from accurately reconstructing the original malware samples based on the compressed ones with the integration of sensors. Last but not least, compared with other classical compressed-sensing-based methods, the proposed method outperforms other methods in terms of malware classification accuracy and image reconstruction quality, either with or without the impacts brought by noises detected by sensors.

It is worth noting that since the application of the proposed method is malware classification, in order to ensure the security of the classification process, in this paper, we mainly focus on imprecise reconfigurability of the proposed method. Actually, a large proportion of compressed-sensing-based image processing methods place great emphasis on the ability of image reconstruction. In the future, other techniques, such as associated learning and training using sensors may be introduced to refine the proposed method in terms of reconstruction performance. In this way, the proposed method may be exploited and applied widely in new fields, such as medical image classification for remote healthcare with the use of advanced sensors.

## Figures and Tables

**Figure 1 sensors-24-04253-f001:**
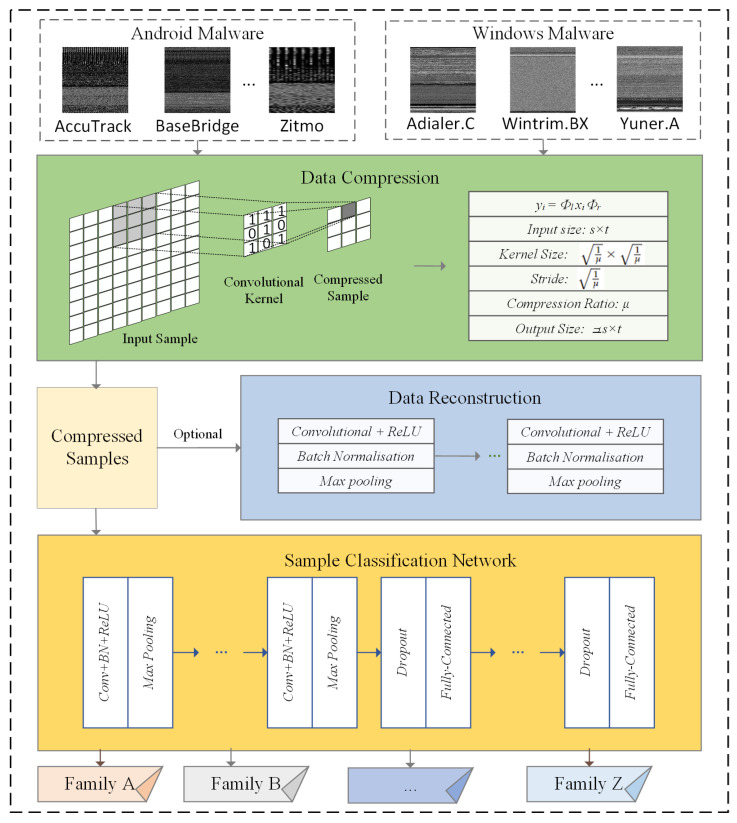
The main workflow of the proposed method. A total of *n* data owners firstly compress their own malware samples, i.e., $x_{i}$, and then share these compressed samples, i.e., $y_{i}$ to accomplish malware classification tasks. Matrices $\Phi_{l}$ and $\Phi_{r}$ are the measurement matrices used by data owners during the data compression process. The compression ratio is μ.

**Figure 2 sensors-24-04253-f002:**
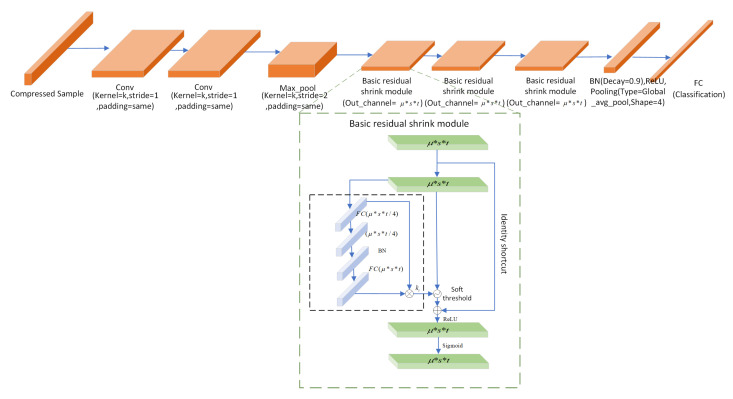
The classification network structure of the proposed method.

**Figure 3 sensors-24-04253-f003:**
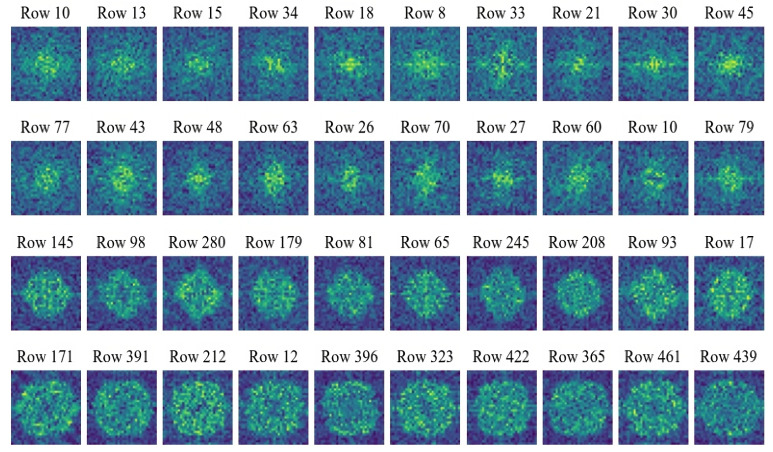
Frequency domain conversion results of the proposed method under various compression ratios. The first to the fourth rows show the results under compression ratios of 0.05, 0.1, 0.3, and 0.5, respectively.

**Figure 4 sensors-24-04253-f004:**
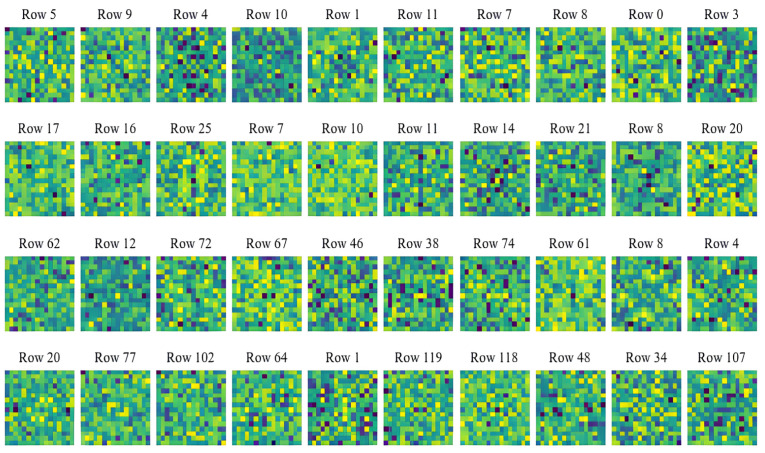
Frequency domain conversion results of compressed sensing method using Gaussian measurement matrix under various compression ratios. The first to the fourth rows show the results under compression ratios of 0.05, 0.1, 0.3, and 0.5, respectively.

**Figure 5 sensors-24-04253-f005:**
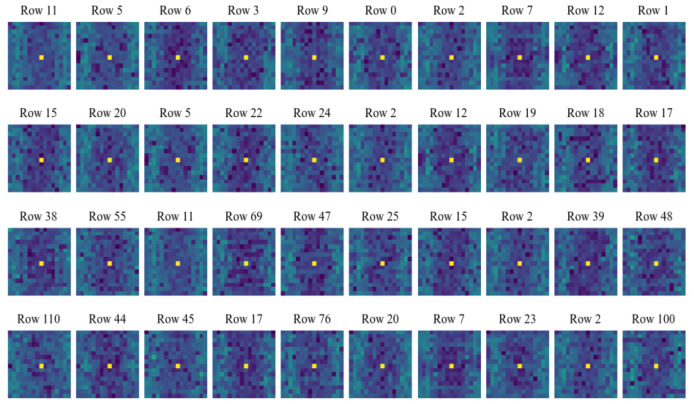
Frequency domain conversion results of compressed sensing method using Logistic measurement matrix under various compression ratios. The first to the fourth rows show the results under compression ratios of 0.05, 0.1, 0.3, and 0.5, respectively.

**Figure 6 sensors-24-04253-f006:**
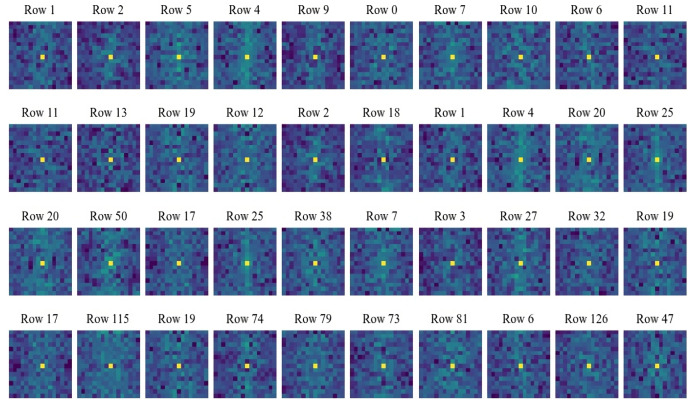
Frequency domain conversion results of compressed sensing method using Tent measurement matrix under various compression ratios. The first to the fourth rows show the results under compression ratios of 0.05, 0.1, 0.3, and 0.5, respectively.

**Figure 7 sensors-24-04253-f007:**
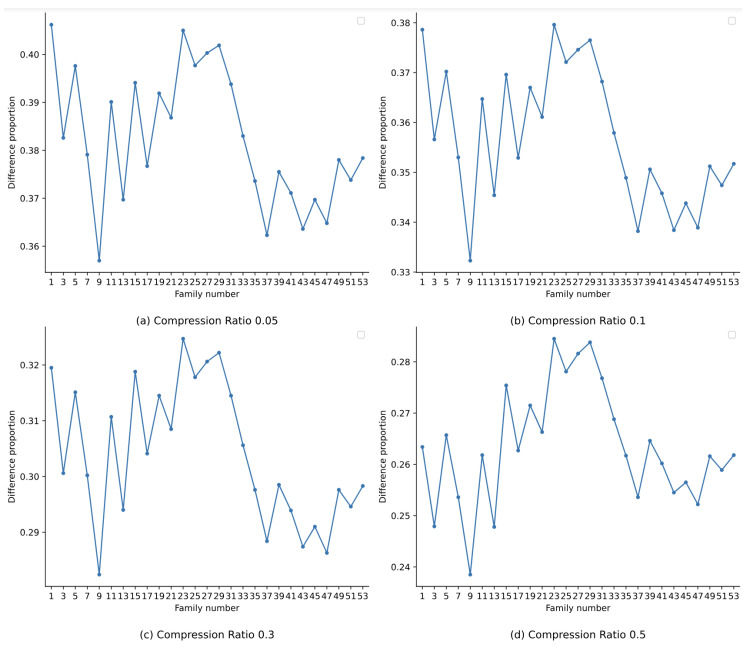
Difference proportion of operation codes for each malware family in Drebin dataset.

**Figure 8 sensors-24-04253-f008:**
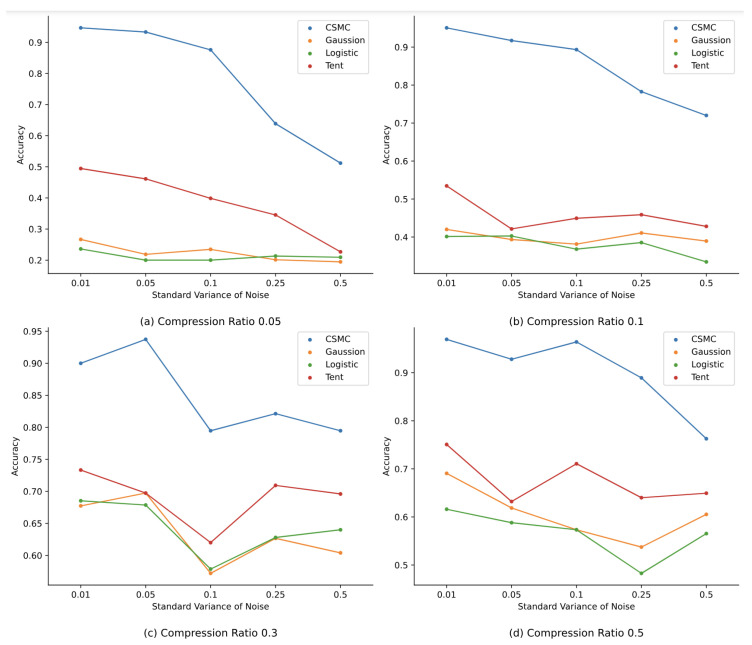
Classification accuracy comparison under the influence of noise.

**Figure 9 sensors-24-04253-f009:**
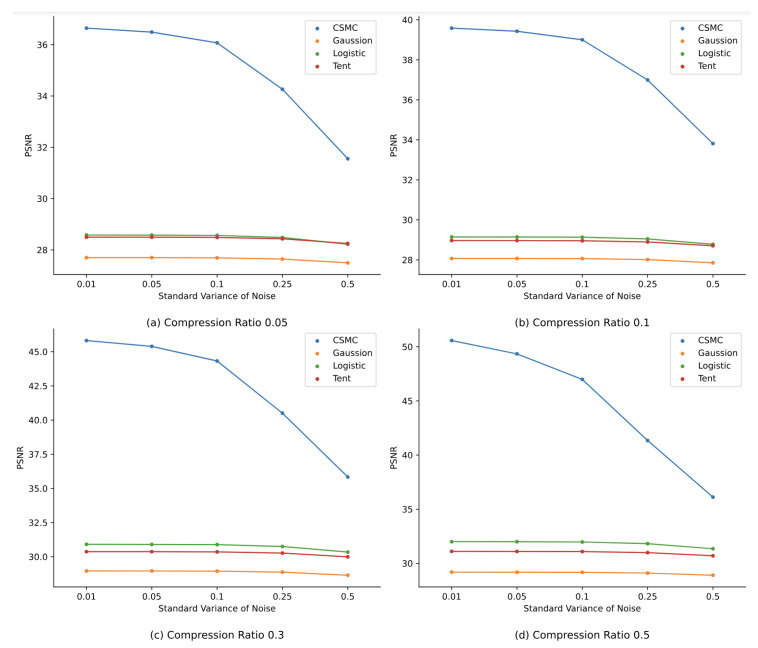
Reconstruction quality comparison under the influence of noise.

**Table 1 sensors-24-04253-t001:** Main structure of the CSMC classification network.

Layer	Channel	Kernel	Stride	Padding	Activation	Annotation
0	c	-	-	-	-	Input Layer
1	c	k = 3×3	1,1	same	Linear	Conv_2d
2	c	k = 3×3	1,1	same	Linear	Conv_2d
3		k = 3×3	2,2	same		Max_pool_2d
4–10	Residual_shrinkage_block (out_channel = μ×s×t)					
11–17	Residual_shrinkage_block (out_channel = μ×s×t×2)					
18–24	Residual_shrinkage_block (out_channel = μ×s×t×2)					
25	Batch_normalization (decay = 0.9)					
26	Activation (ReLU)					
27	Global_avg_pool					
28	Fully_connected (activation = softmax)					
29	Momentum (learning_rate = 0.1, lr_decay = 0.1, decay_step = 20,000)					
30	Regression (loss = categorical crossentropy)					

**Table 2 sensors-24-04253-t002:** Classification and reconstruction performance comparison based on Malimg dataset.

Compression Ratio	Compressed Method	Classification Accuracy	PSNR (dB)
0.05	Gaussian	0.41067	8.05946
	Logistic	0.43733	10.79386
	Tent	0.43999	9.96238
	CSMC	**0.97867**	**19.02935**
0.1	Gaussian	0.57333	8.16376
	Logistic	0.48133	11.08026
	Tent	0.57466	10.18503
	CSMC	**0.98000**	**20.07452**
0.3	Gaussian	0.73333	8.37789
	Logistic	0.74667	11.57817
	Tent	0.73733	10.5878
	CSMC	**0.98000**	**24.32906**
0.5	Gaussian	0.72533	8.50674
	Logistic	0.71733	12.00514
	Tent	0.69067	10.88755
	CSMC	**0.97733**	**28.64138**

**Table 3 sensors-24-04253-t003:** Classification and reconstruction performance comparison based on Drebin dataset.

Compression Ratio	Compressed Method	Classification Accuracy	PSNR (dB)
0.05	Gaussian	0.96852	27.69984
	Logistic	0.98333	28.57814
	Tent	**0.98611**	28.49639
	CSMC	0.97500	**36.65134**
0.1	Gaussian	0.97222	28.07057
	Logistic	0.97685	29.14585
	Tent	0.97222	28.96592
	CSMC	**0.98056**	**39.59088**
0.3	Gaussian	0.97407	28.96270
	Logistic	0.96667	30.90944
	Tent	0.97222	30.37308
	CSMC	**0.98148**	**45.83466**
0.5	Gaussian	0.96759	29.18425
	Logistic	0.97407	32.00142
	Tent	0.97407	31.10516
	CSMC	**0.97778**	**50.63212**

**Table 4 sensors-24-04253-t004:** ANOVA test results.

Metric	F-Value	*p*-Value
Malimg Accuracy	8.450406	0.002744
Drebin Accuracy	1.660423	0.228034
Malimg PSNR	35.495435	0.000003
Drebin PSNR	17.364036	0.000116

**Table 5 sensors-24-04253-t005:** Kruskal–Wallis test results.

Metric	H-Value	*p*-Value
Malimg Accuracy	8.505155	0.036648
Drebin Accuracy	5.379464	0.146028
Malimg PSNR	13.786765	0.003210
Drebin PSNR	10.213235	0.016838

**Table 6 sensors-24-04253-t006:** The classification performance comparison of CSMC and machine or deep learning methods.

Method	Technique	Input	Reconstructable	Accuracy
[42]	Machine Learning	Uncompressed	No	0.9718
[43]	Deep Learning	Uncompressed	No	0.9732
[44]	Machine Learning	Uncompressed	No	0.8911
[1]	Deep Learning	Uncompressed	No	0.9760
[45]	Machine Learning	Uncompressed	No	0.97
[46]	Deep Learning	Uncompressed	No	0.9480
[1]	Machine Learning	Uncompressed	No	0.922
[1]	Machine Learning	Uncompressed	No	0.9190
[1]	Machine Learning	Uncompressed	No	0.9320
[1]	Machine Learning	Uncompressed	No	0.9250
[1]	Deep Learning	Uncompressed	No	0.9450
[47]	Deep Learning	Uncompressed	No	0.976
**This Paper**	Deep Learning	**Compressed**	**Yes**	**0.98**

## Data Availability

Data are contained within the article.

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
