# Peer review of "CSMC: A Secure and Efficient Visualized Malware Classification Method Inspired by Compressed Sensing"

_sensors, 2024, doi:10.3390/s24134253_

Round 1
Reviewer 1 Report
Comments and Suggestions for Authors
The authors articulated their research objective. However, the abstract is too long, so it is suggested that it be formatted according to context, problem statement, methodology, results, and implications.
Additional comments
The original contributions of the paper include:
A novel integration of compressed sensing and deep learning for malware classification.
The focus on secure sharing of malware samples without exposing them to classification participants.
Enhancing robustness against noise in complex network environments, which is particularly relevant for IoT sensors. The specific gap addressed is the need for efficient and secure malware classification methods suitable for resource-constrained IoT environments.
Compared to other published material, this research adds:
An innovative approach that combines compressed sensing and deep learning for malware classification.
A focus on the security aspect of malware sample sharing, which is less explored in existing literature.
Empirical evidence demonstrating superior classification performance on Windows and Android malware samples.
Fig2. Ensure that all components and layers are labeled consistently throughout the figure. For instance, specify the dimensions or types of pooling layers where possible. Figs 3-6. Ensure high-quality images and graphs to enhance the readability and professional presentation of the paper.
End Comments
Reviewer 2 Report
Comments and Suggestions for Authors
1. The introduction does not provide a clear problem statement or research gap that the study aims to address. It would be helpful to explicitly state what specific problem or limitation in current malware classification methods this study seeks to overcome.
2. A few more current references could be added when discussing the state-of-the-art in malware classification to ensure the most up-to-date context is provided. Most of the cited references are from 2017-2019.
3. The related work summarizes various methods but does not provide much direct comparison between them in terms of performance, limitations, datasets used, etc. More explicit analysis of the strengths and weaknesses of each approach would help identify the key gaps that this study aims to address.
4. While some limitations are briefly mentioned (e.g., computational complexity, need for large datasets), a more thorough and consistent analysis of the drawbacks of each cited method would be useful to understand the current challenges in the field.
5. As mentioned for the introduction, adding some more recent references from 2020-2022 would ensure the most current strategies and benchmarks are discussed.
6. There is not much explanation of how key hyperparameters were chosen, such as the number of RSM layers, threshold parameters, learning rate, etc. Discussion of any hyperparameter search or tuning process would be informative.
7. The reported performance improvements are not tested for statistical significance. Conducting appropriate statistical tests would strengthen the reliability of the results.
8. While the feature extraction capabilities are visualized through frequency domain analysis, there is no deeper investigation into what specific features the model is learning and how they relate to malware characteristics. More interpretability analysis could provide useful insights for malware experts.
9. The noise robustness experiments only consider Gaussian noise, but malware data may face other types of corruptions or adversarial perturbations in practice. Testing against a wider range of robustness threats would be valuable.
Round 2
Reviewer 2 Report
Comments and Suggestions for Authors
Thank you for your revised submission. I am pleased to inform you that all my concerns have been addressed satisfactorily. The revisions have significantly improved the manuscript.